# Mechanisms of azole antifungal resistance in clinical isolates of *Candida tropicalis*

**Saikat Paul, Dipika Shaw, Himanshu Joshi, Shreya Singh, Arunaloke Chakrabarti, Shivaprakash M. Rudramurthy**[ID]**, Anup K. Ghosh**[ID]*****

Department of Medical Microbiology, Postgraduate Institute of Medical Education and Research (PGIMER), Chandigarh, India

* anupkg3@gmail.com

**Data Availability Statement:** All relevant data are within the paper and its Supporting Information files.

## Abstract

This study was designed to understand the molecular mechanisms of azole resistance in *Candida tropicalis* using genetic and bioinformatics approaches. Thirty-two azole-resistant and 10 azole-susceptible (S) clinical isolates of *C. tropicalis* were subjected to mutation analysis of the azole target genes including *ERG11*. Inducible expression analysis of 17 other genes potentially associated with azole resistance was also evaluated. Homology modeling and molecular docking analysis were performed to study the effect of amino acid alterations in mediating azole resistance. Of the 32 resistant isolates, 12 (37.5%) showed A395T and C461T mutations in the *ERG11* gene. The mean overexpression of *CDR1*, *CDR3*, *TAC1*, *ERG1*, *ERG2*, *ERG3*, *ERG11*, *UPC2*, and *MKC1* in resistant isolates without mutation (R-WTM) was significantly higher (p<0.05) than those with mutation (R-WM) and the sensitive isolates (3.2–11 vs. 0.2–2.5 and 0.3–2.2 folds, respectively). Although the R-WTM and R-WM had higher (p<0.05) *CDR2* and *MRR1* expression compared to S isolates, noticeable variation was not seen among the other genes. Protein homology modelling and molecular docking revealed that the mutations in the *ERG11* gene were responsible for structural alteration and low binding efficiency between ERG11p and ligands. Isolates with *ERG11* mutations also presented A220C in *ERG1* and together T503C, G751A mutations in *UPC2*. Nonsynonymous mutations in the *ERG11* gene and coordinated overexpression of various genes including different transporters, ergosterol biosynthesis pathway, transcription factors, and stress-responsive genes are associated with azole resistance in clinical isolates of *C. tropicalis*.

## Introduction

*Candida* species, including *Candida albicans* and non-*Candida albicans Candida* (NCAC) species are implicated in a myriad of superficial and invasive infections including bloodstream infections [1]. Morbidity and mortality due to invasive candidiasis (IC) are significantly higher in immunocompromised patients [2–4]. *Candida tropicalis*, among the NCAC species, has emerged as the predominant species responsible for IC in Asian countries including India [3, 5, 6].

**Funding:** We have completed the study as a part of a PhD thesis and by utilizing the institutional research grant (No. 71/2-Edu-16/4856 Dated: 12/12/2017, Budget allotment of Rs. 4,75,000/-). Additionally, the manpower for this study was supported by the Indian Council of Medical Research (ICMR), Government of India. The funders had no role in study design, data collection and analysis, decision to publish, or preparation of the manuscript.

**Competing interests:** The authors have declared that no competing interests exist

**Abbreviations:** NCAC, non-*Candida albicans Candida*; IC, invasive candidiasis; ABC, ATP-binding cassette; MFS, Major facilitator superfamily; YPD, Yeast extract peptone dextrose; BMD, Broth microdilution; CDR, Candida drug resistance gene; MDR, Multi drug resistance gene; ERG1, Squalene epoxidase; ERG2, $\Delta^{8,7}$-isomerase; ERG3, $\Delta^{5,6}$-desaturase; ERG11, Lanosterol C14 alpha-demethylase; ERG24, C14-reductase; HMG, HMG-CoA reductase; MRR1, Multidrug resistance regulator; TAC1, Transcriptional activator of *CDR* genes; UPC2, Transcription factor of *ERG11*; HSP90, heat shock protein 90; HOG1, Mitogen-activated protein kinase involved in osmoregulation; MKC1, Mitogen-activated protein kinase; SOD1, Superoxide dismutase.

Escalating acquired resistance of *C. tropicalis* to currently available antifungal drugs such as azoles derivatives, echinocandins, and amphotericin B has been reported in several studies [1–3, 5, 6]. Factors linked with the development of resistance include, the rampant misuse of anti-fungals, improper dosing resulting in suboptimal drug concentrations, long-term therapy, and the unregulated use of antifungals in agriculture and/or animal husbandry [2, 5]. Fluconazole is perhaps the most commonly used azole because of its low cost, effective bioavailability, and fewer side effects [7, 8]. Azoles act by inhibiting lanosterol C14 alpha-demethylase (ERG11p), an essential enzyme for ergosterol biosynthesis, encoded by the *ERG11* gene.

The predominant mechanism of azole resistance *Candida* species is mutation/ overexpression of different genes [1, 9]. Mutations and overexpression in the azole target *ERG11* are well known mechanisms to be associated with antifungal resistance [10–14], with which the amino acid alterations disrupt the affinity between enzyme and substrate [1, 9, 11, 15–17]. The drug inducible overexpression of transporter genes like the ATP-binding cassette (ABC) and the major facilitator superfamily (MFS) can cause active efflux of cellular azole antifungal drugs thereby contributing to antifungal resistance [1, 9, 17]. Both, the up- and down- regulation of various drug transporters and ergosterol pathway genes have been reported among resistant isolates of *C. tropicalis* [1, 9, 10, 12–14, 16]. There are also reports of alternative mechanisms associated with azole resistance such as mitochondrial defects and biofilm formation [1, 9]. Despite available literature, comprehensive knowledge about the mechanisms behind the azole resistance in *C. tropicalis* is still limited.

The present study was designed to explore the underlying mechanisms of azole resistance in clinical *C. tropicalis* isolates by studying the role of multiple drug transporters, transcription factors, ergosterol biosynthesis pathway, stress-responsive pathways and exploring various known and unknown resistance pathways by using a combination of phenotypic, genetic and bioinformatics approaches.

## Materials and methods

### Yeast isolates

A total of 613 *C. tropicalis* isolates causing invasive infections including 32 azole-resistant isolates were screened for a duration of 4 years (January 2015 to December 2018). These 32 resistant isolates and ten susceptible isolates were included in the present study (Table 1). The isolates examined in this study were also used in our previous studies [18–22]. The institutional ethics committee of the Postgraduate Institute of Medical Education and Research (PGI-MER), Chandigarh, India approved the study. The isolates were identified by MALDI-TOF MS (Microflex LT mass spectrometer, Bruker Daltonik, Germany) and PCR sequencing of the internal transcribed spacer (ITS) region of ribosomal DNA [23, 24].

### Antifungal susceptibility testing (AFST)

Minimum inhibitory concentrations (MICs) with respect to fluconazole, voriconazole, itraconazole, and posaconazole (Sigma-Aldrich, Germany) were assessed for all the isolates using the CLSI broth micro-dilution (BMD) method (M27-A3) followed by MIC interpretation in accordance with the CLSI M27-S4 guidelines [25, 26].

### Sequencing of azole drug target *ERG11*

PCR sequencing of the *ERG11* gene was performed in all the isolates by using two primer pairs as demonstrated in our previous study (S1 Table in S1 File) [20]. SeqMan software (DNAS-TAR, USA) was used to align the multiple fragments of the *ERG11* gene. ClustalX 2.1 software

**Table 1. Details of isolates with clinical background.**

| Isolate ID | Flu MIC (mg/L) | Vori MIC (mg/L) | Itra MIC (mg/L) | Posa MIC (mg/L) | *ERG11* mutations (A395T & C461T) | Amino acid alterations (Y132F & S154F) | Patient Age | Sex | Clinical diagnosis | Type of sample |
|---|---|---|---|---|---|---|---|---|---|---|
| 420182 | 16 | 0.25 | 0.06 | 0.12 | No | No | 69 | M | Perforation peritonitis | Blood |
| 420183 | 64 | 0.25 | 0.25 | 0.12 | No | No | 25 | M | Meningitis | CSF |
| 420184 | 32 | 0.5 | 0.12 | 0.06 | No | No | 60 | M | Sepsis | Blood |
| 420185 | 32 | 2 | 0.25 | 0.06 | No | No | 14 Days | M | Late-onset of neonatal sepsis | Blood |
| 420186 | 16 | 0.12 | 0.12 | 0.06 | No | No | 50 | M | Sepsis | Blood |
| 420187 | 32 | 1 | 0.06 | 0.06 | No | No | 29 | M | Acute Chronic Liver failure | Blood |
| 420188 | 16 | 0.5 | 0.03 | 0.06 | No | No | 22 | M | Meningioma | Blood |
| 420189 | 128 | 4 | 0.5 | 0.5 | Yes | Yes | 67 | M | Sepsis | Blood |
| 420190 | 16 | 0.25 | 0.06 | 0.03 | No | No | 35 | M | Burns | Blood |
| 420201 | 64 | 0.5 | 0.06 | 0.06 | No | No | 54 | M | Sepsis | Blood |
| 420191 | 64 | 0.25 | 0.06 | 0.06 | No | No | 67 | M | Shock | Blood |
| 420192 | 16 | 0.25 | 0.03 | 0.06 | No | No | 60 | M | Pancreatitis | Blood |
| 420193 | 128 | 1 | 0.12 | 0.25 | No | No | 29 | M | Poisoning | Blood |
| 420194 | 32 | 0.25 | 0.03 | 0.06 | No | No | 50 | M | Respiratory Distress | Blood |
| 420195 | 128 | 4 | 2 | 1 | No | No | 29 | M | Pancreatitis | Blood |
| 420227 | 128 | 0.5 | 0.25 | 0.5 | Yes | Yes | 45 | F | Pancreatitis | Pus |
| 420228 | 256 | 4 | 2 | 2 | No | No | 58 | M | Septic shock | Blood |
| 420229 | 128 | 4 | 2 | 2 | No | No | 35 | M | Lung Carcinoma | Blood |
| 420230 | 256 | 4 | 2 | 2 | No | No | 10 | M | Sepsis | Blood |
| 420231 | 256 | 2 | 0.12 | 0.12 | No | No | 73 | M | Septic shock | CSF |
| 420232 | 32 | 0.5 | 0.5 | 0.5 | Yes | Yes | 60 | F | Roadside accident | Blood |
| 420233 | 32 | 1 | 0.25 | 0.25 | Yes | Yes | 20 | M | Pancreatic injury | Blood |
| 420234 | 64 | 1 | 0.25 | 0.25 | Yes | Yes | 50 | M | Sepsis | Blood |
| 420235 | 32 | 0.5 | 0.25 | 0.25 | Yes | Yes | 20 | M | Gastric perforation peritonitis | Blood |
| 420236 | 32 | 0.5 | 0.25 | 0.25 | Yes | Yes | 20 | M | Gastric perforation peritonitis | Blood |
| 420237 | 64 | 1 | 0.5 | 0.5 | Yes | Yes | 50 | M | Sepsis | Blood |
| 420238 | 256 | 16 | 16 | 2 | Yes | Yes | 20 | M | Sepsis | Blood |
| 420239 | 256 | 16 | 16 | 0.5 | Yes | Yes | 14 Days | M | Seizure | Blood |
| 420245 | 128 | 2 | 1 | 0.5 | Yes | Yes | 14 Days | M | Jejunal atresia | Blood |
| 420246 | 32 | 1 | 0.25 | 0.5 | No | No | 1 Month | F | Meningitis | Blood |
| 420247 | 128 | 4 | 2 | 0.25 | Yes | Yes | 28 | M | Leg fracture | Wound slough |
| 420248 | 16 | 0.5 | 0.25 | 0.25 | No | No | 14 Days | F | Tracheoesophageal fistula | Blood |
| 420214 | 1 | 0.03 | 0.06 | 0.06 | No | No | 2 months | F | Sepsis | Blood |
| 420215 | 0.5 | 0.06 | 0.12 | 0.03 | No | No | 77 | M | Post-op gastrectomy | Blood |
| 420203 | 1 | 0.12 | 0.12 | 0.06 | No | No | 84 | F | Cerebral venous accident | Blood |
| 420200 | 0.5 | 0.03 | 0.03 | 0.06 | No | No | 52 | F | Sepsis | Blood |
| 420212 | 0.5 | 0.25 | 0.12 | 0.25 | No | No | 32 | M | Sepsis | Blood |
| 420210 | 0.5 | 0.03 | 0.06 | 0.06 | No | No | 62 | F | Ovarian carcinoma | Blood |
| 420199 | 1 | 0.03 | 0.12 | 0.12 | No | No | 23 | M | Road traffic accident | Blood |
| 420205 | 1 | 0.25 | 0.12 | 0.06 | No | No | 65 | M | Extrahepatic Biliary obstruction | Ascitic Fluid |

*(Continued)*

**Table 1.** (Continued)

| Isolate ID | Flu MIC (mg/L) | Vori MIC (mg/L) | Itra MIC (mg/L) | Posa MIC (mg/L) | *ERG11* mutations (A395T & C461T) | Amino acid alterations (Y132F & S154F | Patient Age | Sex | Clinical diagnosis | Type of sample |
|---|---|---|---|---|---|---|---|---|---|---|
| 420204 | 0.5 | 0.06 | 0.12 | 0.03 | No | No | 8 | M | Anemia decreased evaluation | Blood |
| 420198 | 0.5 | 0.12 | 0.06 | 0.03 | No | No | 28 | M | Sepsis | Blood |

Flu: Fluconazole; Vori: Voriconazole; Itra: Itraconazole; Posa: Posaconazole; CSF: Cerebrospinal fluid; A: Adenine; T: Thymine; C: Cytosine; Y: Tyrosine; F: Phenylalanine; S: Serine

(UCD Conway Institute, Ireland) was used to align the consensus sequence of the isolates with respect to *C. tropicalis* MYA- 3404 (GenBank accession no. XM_002550939.1) to determine the molecular alterations.

## Expression analysis of target genes

Drug-induced expression of 17 genes [ergosterol synthesis genes (*ERG1*, *ERG2*, *ERG3*, *ERG11*, *ERG24*, and *HMG*), drug efflux transporter genes (*CDR1*, *CDR2*, *CDR3*, and *MDR1*), transcription factors [Multidrug resistance regulator (*MRR1)*, Transcriptional activator of *CDR* genes (*TAC1*) and Transcription factor of *ERG11* (*UPC2*)], and different stress pathway genes (*HSP90*, *HOG1*, *MKC1*, and *SOD1*)] was studied. The primers from our previously published study were used for expression analysis (S2 Table in S1 File) [20]. The RT-qPCR based expression analysis was performed as described previously [20, 21]. In brief, after the incubation of cells for 7 hours with and without drug, total RNA was extracted with TRIzol reagent (Invitrogen, California, USA). The quality and quantity of the RNA was confirmed by NanoDrop (Thermo Scientific, Massachusetts, USA) and the 260/280 for the samples was in between 1.85 and 2.1. The cDNA was synthesized using High-capacity cDNA synthesis kit (Thermo Fisher Scientific, Massachusetts, USA) with 1μg RNA input and Eppendorf 5331 MasterCycler (Eppendorf, Hamburg, Germany) was used for amplification. Expression of the target genes were examined with the Light Cycler 480 (Roche, Switzerland) RT-qPCR system using the PowerUp SYBR Green Master Mix (Termo Fisher Scientific, United States) following the manufacturer's instructions using 1μL cDNA. The RT-qPCR running protocol was as follows: One cycle initial denaturation at 95°C for 1 minute; 45 repeated cycles of denaturation, annealing and extension at 94°C for 10 seconds, 59°C for 10 seconds, 72°C for 10 seconds respectively. Finally, melting curve was generated using the setup at 95°C for 5 seconds, 59°C for 1 min and 97°C for 15 seconds. The expression of the genes was analyzed with respect to untreated control using the ΔΔCT method [27]. We optimized elongation factor 1α (*EF1*) as the stable reference gene and used for the drug-induced expression of the target genes [21].

## Homology modelling and model quality assessment

The model of both wild and mutant protein was generated by different programs and model quality scores were analyzed (S3 Table in S1 File). To investigate the structural variations upon mutation, structural superimposition of both wild and mutant types was performed. ΔΔG value (Gibbs free energy) was calculated to infer the effect of mutations on the structural stability of the protein. The detailed methodology of homology modelling and model quality assessment is explained in the S1, S2 and S4 Material and Methods in S1 File.

## Molecular docking study

Docking analysis was performed to determine the binding affinity of fluconazole and voriconazole against the lanosterol 14-alpha demethylase (ERG11p) of both wild and mutant *C. tropicalis*. After every successful docking simulation, the model falling in the top-ranked cluster with the strongest binding energy was utilized for further analysis. The methodology of molecular doc king is explained in the S3 and S4 Material and Methods in S1 File.

## Sequence analysis of other resistance-related genes

Apart from the *ERG11*, sequencing of *ERG1*, *ERG3*, *UPC2*, and *TAC1* genes was performed as these genes are well documented to be associated with azole resistance in *Candida*. An attempt was taken to analyze the nonsynonymous mutations among these genes. The gene sequences of *C. tropicalis* MYA-3404 was used as a reference for mutation analysis of target genes in isolates used in this study.

## Statistical analysis

GraphPad software (GraphPad Prism 9, California, USA) was used for statistical analysis. Statistical significance was computed using Kruskal-Wallis test, Student's t-test and ANOVA. A p-value <0.05 is significant.

# Results

## Clinical details of the isolates

The 42 isolates used in the present study, were recovered from blood, ascitic fluid, cerebrospinal fluid, pus, and wound slough. Out of 32 azole-resistant *C. tropicalis*, 28 (87.5%) patients were male and 4 (12.5%) were female. Most of the patients were ≥ 20 years old and presented a huge diversity in the underlying conditions is present. Among the patients, 8 (25%) patients were receiving fluconazole treatment for 7–28 days. Among the 10 susceptible isolates, 3 were exposed to fluconazole for 7–14 days (Table 1).

## Antifungal susceptibility profile and nonsynonymous mutations in the *ERG11* gene

All the 32 fluconazole-resistant isolates showed the MICs between 16 to 256 mg/L, while in the 10 fluconazole susceptible isolates MICs ranged between 0.5 to 1mg/L. Out of 32 fluconazole-resistant isolates cross-resistance to voriconazole, itraconazole, and posaconazole was presented by 17(1–16 mg/L), 8(1–16 mg/L), and 5(1–2mg/L) respectively (Table 1).

Out of the 32 resistant isolates, *ERG11* mutations at 395 and 461 positions were observed in 12 (37.5%) isolates. At 395 position, adenine (A) was replaced by thymine (T) whereas, at 461 position, cytosine (C) was replaced by T. Due to these two alterations, Tyrosine (Y) to Phenylalanine (F) substitution at 132 position and Serine (S) to F alteration at 154 position was seen in the protein sequence of Lanosterol 14-alpha demethylase enzyme (ERG11p). No nonsynonymous mutations were noticed among the susceptible isolates (Table 1).

## Inducible expression of resistance related genes

To determine the inducible expression of the genes, freshly frown cells at a concentration of $1 \times 10^6$ cells/mL were inoculated in Yeast extract peptone dextrose (YPD) broth. After 4 hours, cells were treated with sub-inhibitory concentration of fluconazole (Two dilutions lower than the MIC of the isolates) along with another setup with untreated control. Cells were incubated

up to 7 hours and the expression of the different genes were analyzed. The mean inducible expression of *CDR1*, *CDR3*, and *TAC1* was significantly higher (p<0.05) in the 20 resistant isolates without *ERG11* mutations (R-WTM) at 4.9, 4.5, and 3.2 folds respectively compared to the 12 resistant isolates with *ERG11* mutations (R-WM) at 1.8, 1.6, and 2 folds respectively and the 10 susceptible isolates (S) at 0.3, 1, and 1.4 fold respectively. On the other hand, expression of *CDR2* and *MRR1* in R-WTM (2.1 and 1.8 fold respectively) and R-WM (2.2 and 1 fold respectively) was significantly higher (p<0.05) than the S isolates (0.02 and 0.1 fold respectively). No significant variation (p>0.05) in the *MDR1* expression was noted in the R-WTM, R-WM, and S isolates (Fig 1A–1F).

The average fold overexpression of *ERG1*, *ERG2*, *ERG3*, *ERG11*, and *UPC2* in R-WTM (11, 3.4, 5, 6.1, and 4.6 respectively) was significantly (p<0.05) higher than the R-WM (1.5, 1.4, 1.6, 2.5, and 2 respectively) and S (1.3, 2, 2.2, 2.2, and 0.3 respectively) isolates. Though the mean *ERG24* expression was comparatively higher in resistant isolates compared to S isolates, no

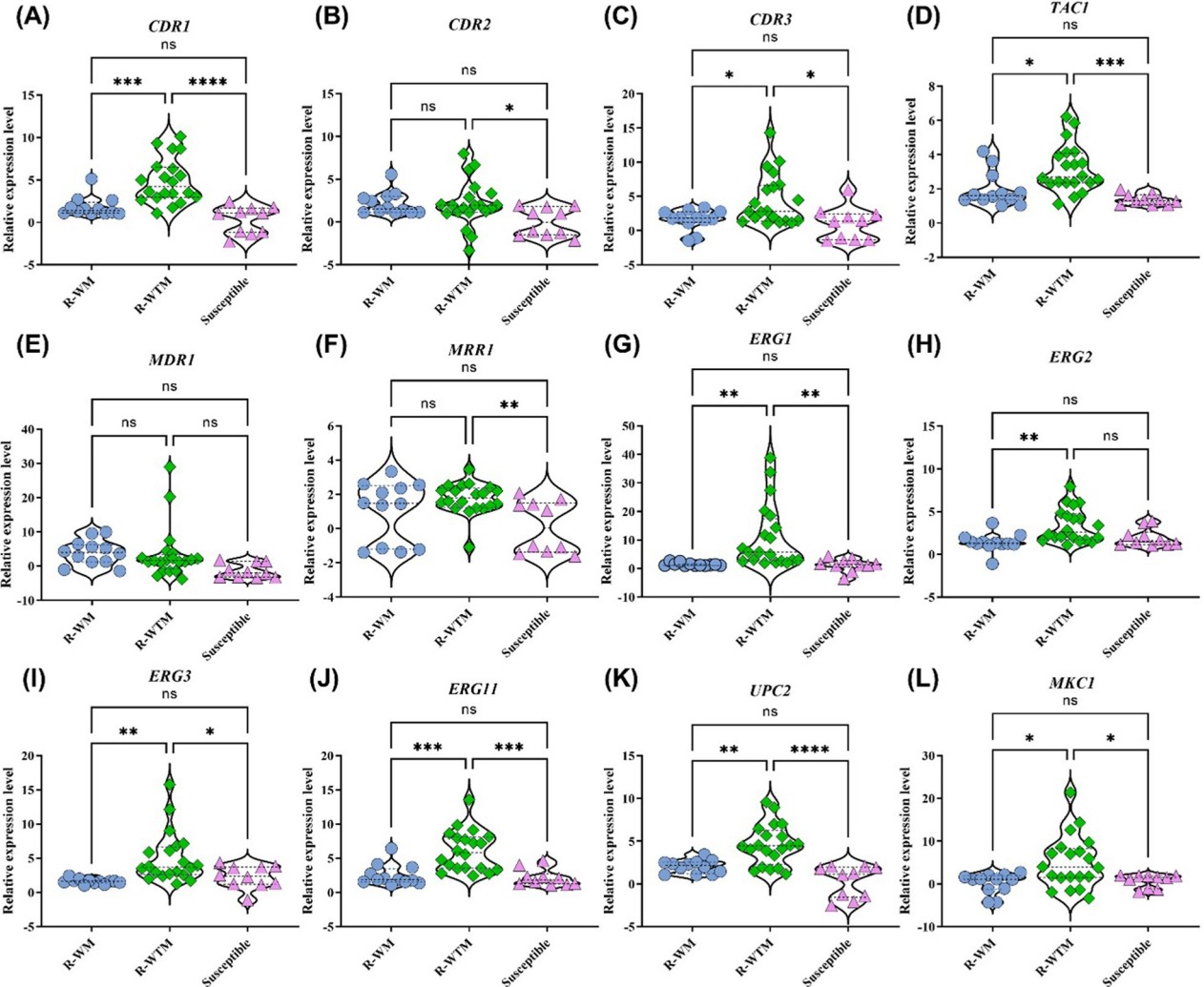

**Fig 1. Scatter dot plots depicting the inducible expression of different transporters (*CDR1*, *CDR2*, *CDR3* and *MDR1*), ergosterol biosynthesis pathway genes (*ERG1*, *ERG2*, *ERG3*, *ERG11*, and *ERG24*), and transcription factors (*TAC1*, *MRR1* and *UPC2*) represented as fold change relative to untreated control.** The level of expression was calculated using $2^{-\Delta\Delta CT}$ method. One-way ANOVA with multiple comparisons was perform to determine the statistical significance. * p<0.05, ** p<0.01, *** p<0.001, **** p<0.0001, and NS = Non Significant.

statistically significant difference was seen in the mean *ERG24* and *HMG* expression among the R-WTM (4 and 0.1), R-WM (2.5 and 0.4), and S (2.7 and 0.5) isolates (p>0.05) (Fig 1G–1K and S1 Fig in S1 File).

No noticeable variation in the expression of *HSP90*, *HOG*, and *SOD1* was seen among R-WTM, R-WM, and S isolates (p>0.05). However, the *MKC1* expression was significantly higher in R-WTM (5.1 fold) compared to both the R-WM (0.2 fold) and S (0.8 fold) isolates (p<0.05) (Fig 1L and S2 Fig in S1 File).

The double gradient heat map confirmed that the ergosterol biosynthesis pathway genes are activated in all three groups. The level of their expression in azole-resistant isolates was higher compared to susceptible isolates. In contrast, a higher level of transporter gene expressions was only noted in resistant isolates. Among the stress-responsive genes, *MKC1* expression was only observed in R-WTM. All together the ergosterol biosynthesis pathway genes, transporter genes, and stress-responsive genes are coordinately expressed specifically in resistant isolates (Fig 2). The heat map is confirming a probable interrelation between the genes and their direct effect on azole resistance.

### *ERG11* expression in cross-resistance isolates

The overexpression of azole drug target *ERG11* gene was also measured among isolates resistant to only fluconazole (Flu), cross-resistant to Flu and voriconazole (Flu+Vori) and also cross-resistant to Flu, Vori, and itraconazole (Flu+Vori+Itra). No significant variation in average fold expression levels (4.8, 5.4, and 3.9 respectively) was seen among these three groups (p>0.05) (S3 Fig in S1 File).

### Homology modelling of lanosterol 14-alpha demethylase (ERG11p)

BLASTp search against the PDB database shows that the protein has 83.11% identity with protein Lanosterol C14 alpha demethylase (PDB ID: 5V5Z) from *Candida albicans*. The sequence had an overall query coverage of 99% and 91% similar amino acids. Hence, the 5V5Z was selected as template for the tertiary structure predication. Modeller v9.25 was used for homology modelling and the best model was selected based on the minimum DOPE score generated. The model was further subjected to energy minimization using Swiss Pdb viewer and Chimera. After each minimization, the structure was verified using SAVES server. Mutagenesis was achieved using Pymol and the structure was also subjected to refinement procedures same as wild type structure. Gibbs free energy calculation ($\Delta\Delta$G for Y132F = 5.17, S154F = 9.27 and overall $\Delta\Delta$G = 8.74) suggested that the reported mutations are destabilizing the protein (Fig 3).

### Molecular docking

After each successful docking simulation, the pose that falls in the top-ranked cluster with highest binding energy was used for post docking analysis. Results from the docking study reflected that binding energy of native protein is low compared to mutated protein. Binding energy of fluconazole against the native protein was -6.83 kcal/mol [(Fig 4(A1)]; whereas binding energy of fluconazole against the mutant protein was -6.38 kcal/mol [(Fig 4(A2)]. Similarly binding energy of voriconazole against the native protein was -7.44 kcal/mol [(Fig 4(B1)] and -7.22 kcal/mol [(Fig 4(B2)] against the mutant protein. The potential binding site analysis revealed that Tyrosine 132 is highly crucial in forming hydrogen bonds between heme cofactor and the drug molecules i.e fluconazole and voriconazole in the native form. Substitution of Tyrosine 132 by Phenylalanine 132 negates the hydrogen bond between both cofactor and the ligand molecule (Fig 4).

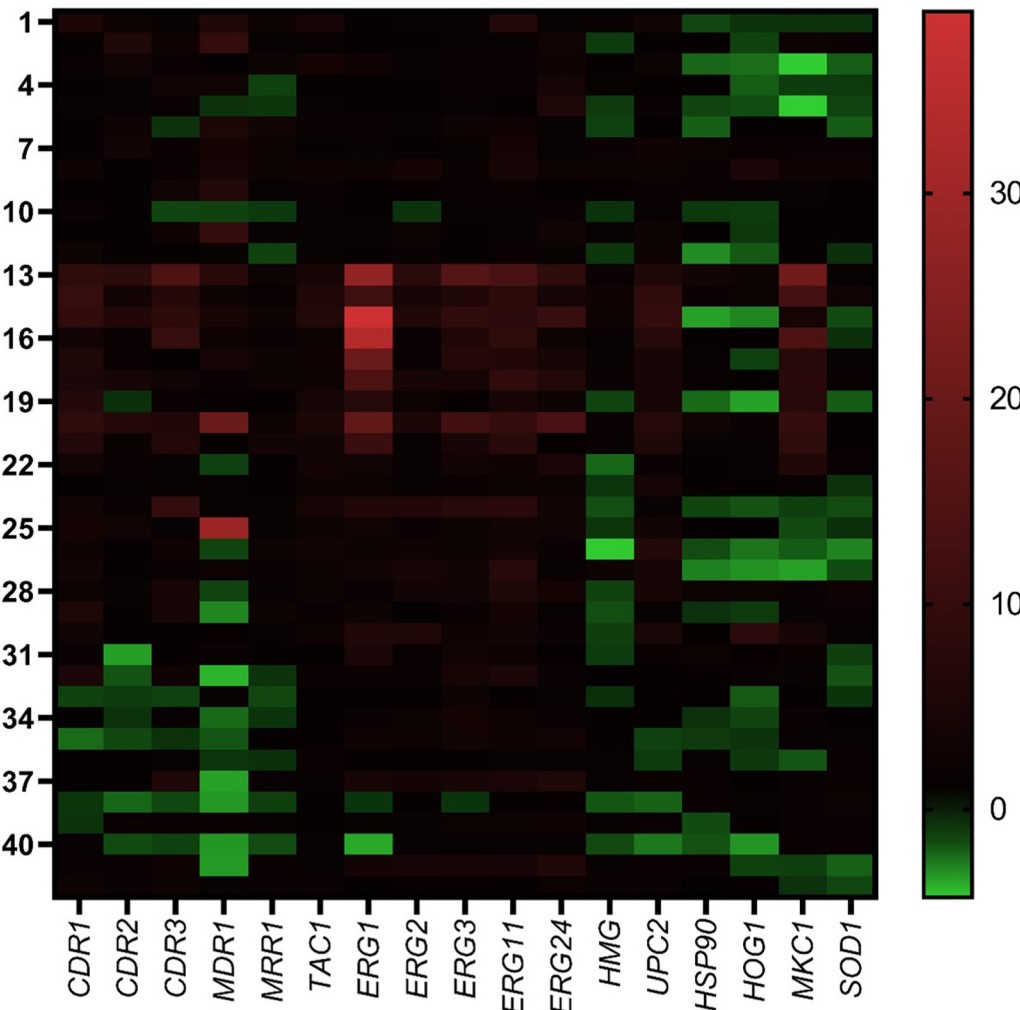

**Fig 2. Heat map demonstrating the comparison between the inducible expression of azole resistance genes among the R-WM (Isolate 1–12), R-WTM (13–32), and S (33–42) isolates.** 'Y' axis is representing the isolates used and 'X' axis representing the genes tested. The scale representing the upregulation (in red) and downregulation (in green) of the genes among resistant and susceptible isolates.

### Analysis of additional resistance-related genes

Surprisingly, the 12 isolates (R-WM) presented *ERG11* mutations also presented nonsynonymous mutations in the coding sequences of *ERG1* and *UPC2* genes. Asparagine (N) to Histidine (H) substitution at 74 position in the Squalene epoxidase enzyme (*ERG1p*) was noted due to A220C transversion in the *ERG1* gene. The T503C and G751A mutations in *UPC2* transcription factor were responsible for Leucine (L) to Proline (P) and Alanine (A) to Threonine (T) substitution at 168 and 251 positions (S4 Table in S1 File). No nonsynonymous mutation was seen in the coding sequences of *ERG3* and *TAC1* genes. We could not build the model of ERG1p and UPC2p due to the lack of a proper template.

## Discussion

A paradigm shift in the epidemiology of IC with an increase in reports of *C. tropicalis* infections and rising azole resistance has been reported in Asian countries including India [3, 5, 6].

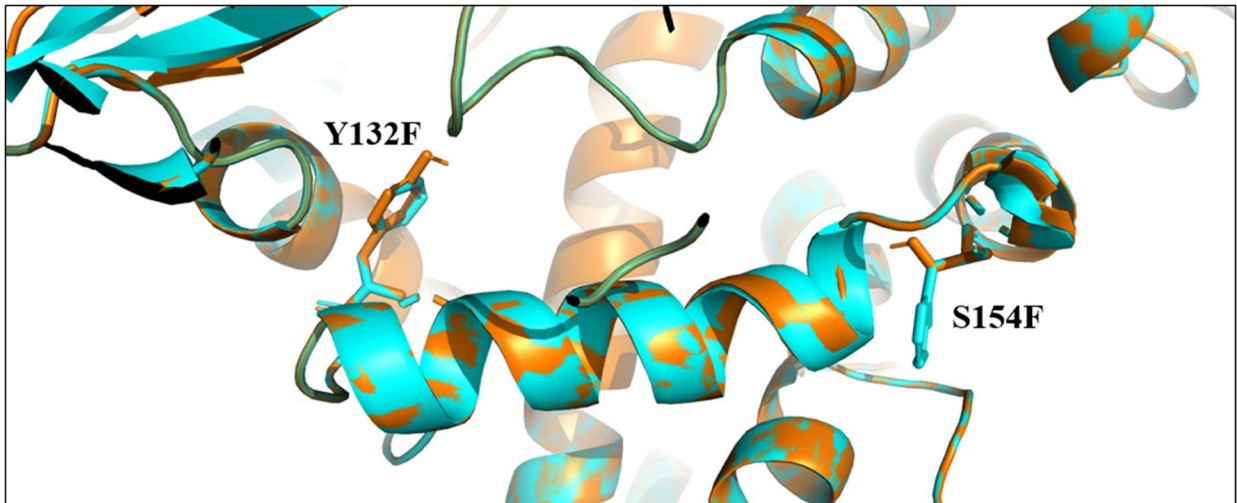

**Fig 3. Homology modelling of ERG11p.** Structural superimposition of both wild and mutant type. Wild type is colored in orange and mutant is in cyan. Mutated residues are shown in stick representation and labelled accordingly.

The mechanism of azole resistance was investigated in the present study among 32 clinical isolates of *C. tropicalis* primarily with respect to their drug efflux transporters, azole antifungal drug target *ERG11* and other ergosterol biosynthesis pathway genes, different transcription factors and stress pathways.

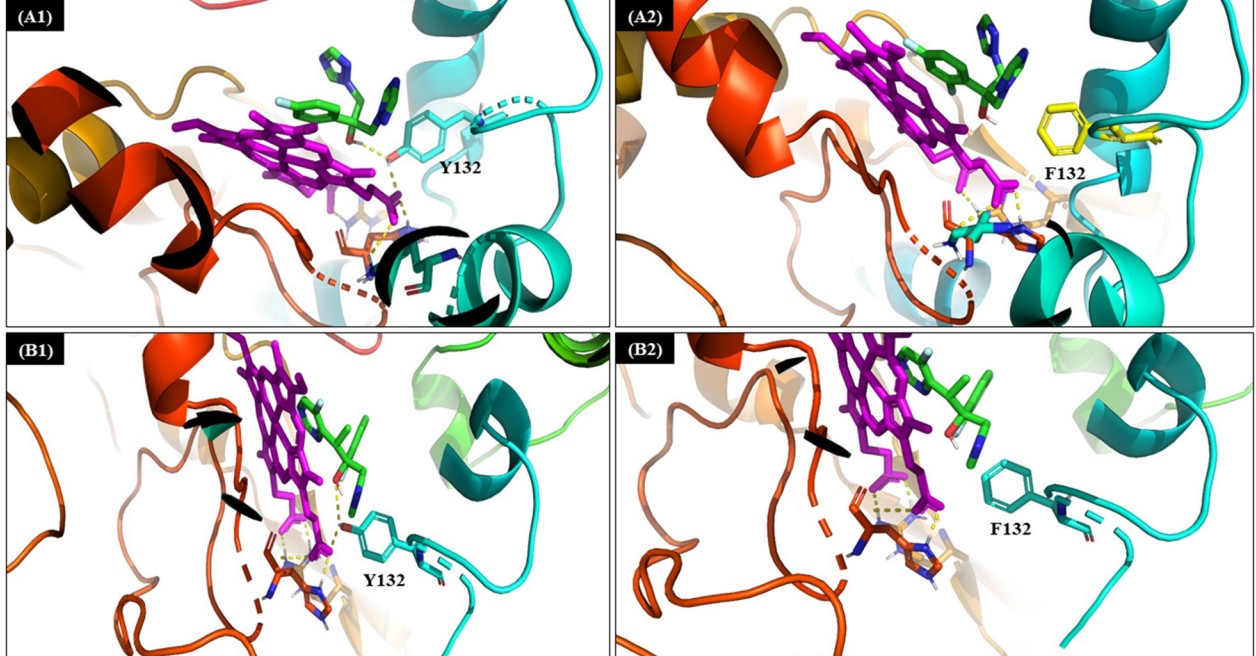

**Fig 4.** Docked pose and interacting residues of (A1) wild protein with Tyrosine132 (cyan) H-binding to Heme (purple) and Fluconazole (green) (A2) mutated protein with Phenylalanine 132 (yellow) in presence of Heme (purple) and fluconazole (green) displays no H-bonding (B1) wild protein with H-bonding of Tyrosine 132 with Heme (purple) and Voriconazole (green) (B2) mutated protein with no H-bonding of Phenylalanine in presence of Heme (purple) and Voriconazole (green). For clarity, only selected binding site residues are shown.

Various host, drugs, and microbial factors are associated with resistance [2]. From the age, sex and background conditions of the patients it is inconclusive whether the host factors are actively responsible for the development of azole resistance or not. Further studies with large number of azole resistant *C. tropicalis* clinical isolates are needed for further conclusion. Of our 32 resistant *C. tropicalis* isolates, only 8 (25%) were under fluconazole treatment for <1 month, suggesting the possible role of other biotic and abiotic factors in the development of azole resistance or the probable acquisition of infection from some unknown sources. Unfortunately, the complete clinical details from the patients could not be retrieved for analysis in the context of their clinical data. Among all *C. tropicalis* isolates in the study period, 5.2% (32 of 613) were fluconazole-resistant with the presence of cross-resistance to voriconazole 53.1% (17 of 32), itraconazole 25% (8 of 32) and even posaconazole 15.6% (5 of 32) indicating that other azoles may not be effective in the scenario.

The active role of mutations in the *ERG11* gene is reported to be responsible for azole resistance in *C. tropicalis* [10–14]. In the present study, 12(37.5%) resistant isolates (MICs: 32–256 mg/L) showed two non-synonymous mutations (A395T and C461T) in the coding sequence of the *ERG11* gene indicating their possible role in mediating azole resistance in *C. tropicalis*. The study by Fan et al. reported that the *C. tropicalis* isolates with non-wild-type *ERG11* gene presented higher MICs; however, we didn't find any noticeable variation among R-WM and R-WTM groups [14].

Overexpression of *ERG11* and different drug efflux transporters (*CDR* and *MDR*) is a well-documented mechanism of resistance in *C. tropicalis* [1, 9, 13, 14]. Most reports suggest no difference in their expression between azole-resistant and susceptible isolates of *C. tropicalis* [10, 28]. However, in our study, while the expression of *MDR1* was uniformly low in both resistant and susceptible isolates, expression of *ERG11*, *CDR1*, and *CDR3* was significantly higher in R-WTM compared to S isolates, indicating their active role in azole resistance. Although, the role of other ergosterol biosynthesis pathway genes in azole resistance have also been demonstrated in previous studies, the significantly higher expression of *ERG1*, *ERG2*, and *ERG3* in resistant isolates is reported in the present study for the first time [29, 30]. Among the transcription factors, Jiang C et al. has explored the role of only *UPC2* on azole resistance in *C. tropicalis* [31]. Here we explored the role of *MRR1* and *TAC1* in addition to *UPC2* in *C. tropicalis* azole resistance. Our result showed a significantly higher level of *TAC1*, *UPC2*, and *MRR1* expression in R-WTM isolates which affects the regulation of their target genes. While previous studies have investigated the role of *SOD1* in azole-resistant *C. tropicalis*, we have additionally examined the expression of other stress pathway genes like *HSP90*, *HOG1*, and *MKC1* for the first time [32]. Among these genes, only *MKC1* presented a higher level of expression among R-WTM, indicting its probable role in mediating azole resistance. We also studied the expression of *ERG11* in azole resistant and cross-resistant isolates; however, no difference in expression was seen. Our previous published work on the experimentally induced fluconazole resistance in *C. tropicalis* supports the findings of this study [20].

Several studies have reported that Y132F and S154F polymorphisms due to alterations at 395 and 461 positions of *C. tropicalis* ERG11p made the target enzyme resistant to azoles [10, 12–14] and these alterations were seen in the 12 resistant (R-WM) isolates even in the present study. These two mutations appeared together consistently, which is similar to the findings of Jiang et al. [10]. Homology modelling analysis in the past has revealed that tyrosine to phenylalanine conversion at 132 position in ERG11p is responsible for the loss of the normal hydrogen bonding between tyrosine and heme. Since heme is an important cofactor required for binding of azole to ERG11p, this alteration affects drug binding [11, 16]. In the present study, the ΔΔG revealed, these two mutations probably destabilized the protein. Tan et al. speculated, Y132F alteration in ERG11p would reduce the affinity between the target site of enzyme and

fluconazole, as it is a hydrophilic drug molecule [11]. It is known that, tyrosine is an aromatic amino acid and is preferentially substituted by amino acids with similar properties and considering the fact that phenylalanine is a precursor of tyrosine and it is a likely substitution in the ERG11p. We confirmed for the first time that substitutions of Y132F and S154F hindered the formation of Pi-Pi and Pi-cation interaction between the cofactor and the ligand molecules thereby dipping the overall binding energy of the mutated docked complex. Moreover, Y132F within the active site greatly altered the hydrophobicity and overall geometry of the active site. Therefore, these two amino acid substitutions reduced the binding energy of both the ligand molecules and in turn conferred resistance towards the ligand molecule.

Previous studies have demonstrated the role of *ERG1* and *UPC2* mutations in azole resistance [13, 31, 33, 34]. Tsai et al. reported that *ERG1* mutation in *C. glabrata* increased the susceptibility to azoles [33]. However, nonsynonymous *ERG1* mutation (A220C) was seen concomitant with *ERG11* mutation among resistant isolates in our study. Gain-of-function (GOF) mutations in the transcription factor *UPC2* have been reported to transform it into a hyperactive state, responsible for azole resistance [31, 34]. Similar to the previous studies, two nonsynonymous mutations in *UPC2* were seen in the present study [13, 35], while the mutation at 503 position is novel. Due to the unavailability of the proper template, the homology modeling of ERG1p and UPC2p could not be performed. Although mutations in *ERG3* and *TAC1* genes have been reported in azole-resistant isolates, we didn't notice any alteration in the coding sequences of these genes [12, 36]. Further studies are needed for functional validation of our findings, preferably in a larger number of isolates to determine the exact role of ERG1p and UPC2p in azole-resistant *C. tropicalis*. Detailed analysis of our findings in the context of clinical information such as duration of hospitalization, prior antifungal treatment and duration, other antimicrobial therapies etc., could also help in providing a better understanding of the factors driving the development of infections due to resistant isolates.

## Conclusions

In conclusion, nonsynonymous mutations in the *ERG11* gene were one of the predominant mechanisms of azole resistance in clinical isolates of *C. tropicalis* as demonstrated by molecular docking analysis for the first time. In addition, to studying the overexpression of previously known genes, we demonstrated the involvement of different transporters, ergosterol biosynthesis pathway genes, transcription factors, and stress pathway genes in azole resistance. However, in view of the rising azole resistance in *C. tropicalis* clinical isolates, systematic and extensive future studies are essential to fully elucidate the mechanisms driving resistance.

## Supporting information

**S1 File.**
(DOCX)

## Acknowledgments

The authors are thankful to the Indian Council of Medical Research (ICMR), Government of India for financial supports. We express our gratitude to the Department of Medical Microbiology, PGIMER, Chandigarh for allowing us to conduct this study.

## Author Contributions

**Conceptualization:** Saikat Paul, Anup K. Ghosh.

**Data curation:** Saikat Paul, Dipika Shaw, Himanshu Joshi.

**Formal analysis:** Saikat Paul, Dipika Shaw, Himanshu Joshi, Anup K. Ghosh.

**Investigation:** Saikat Paul, Shreya Singh, Anup K. Ghosh.

**Methodology:** Saikat Paul, Dipika Shaw, Himanshu Joshi, Anup K. Ghosh.

**Project administration:** Anup K. Ghosh.

**Resources:** Saikat Paul, Anup K. Ghosh.

**Software:** Saikat Paul, Dipika Shaw, Himanshu Joshi, Anup K. Ghosh.

**Supervision:** Shreya Singh, Arunaloke Chakrabarti, Shivaprakash M. Rudramurthy, Anup K. Ghosh.

**Validation:** Saikat Paul, Himanshu Joshi, Anup K. Ghosh.

**Visualization:** Saikat Paul, Anup K. Ghosh.

**Writing – original draft:** Saikat Paul.

**Writing – review & editing:** Saikat Paul, Shreya Singh, Arunaloke Chakrabarti, Shivaprakash M. Rudramurthy, Anup K. Ghosh.

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
