## [Decision Letter · Decision Letter 0]

31 Jan 2022

PONE-D-21-29687Mechanisms of azole antifungal resistance in clinical isolates of Candida tropicalisPLOS ONE

Dear Dr. Ghosh,

Thank you for submitting your manuscript to PLOS ONE. After careful consideration, we feel that it has merit but does not fully meet PLOS ONE’s publication criteria as it currently stands. Therefore, we invite you to submit a revised version of the manuscript that addresses the points raised during the review process.

We are sorry for the late response but it was difficult to get reviewers. The manuscript contains some already reported content but also some new information which may be of interest to the Candida community. There are some issues that need to be addressed.The experimental parameters need to be stated and more clearly described.Relative expression levels need to be readdressed as discussed by reviewer #2Review the discussion and conclusions to confirm they reflect results (see reviewer #2) and also identify content that has already been reported (reviewer #1).Please consider all the comments of the reviewers and if you think their suggestions would improve what you are trying to emphasize.

We look forward to receiving your revised manuscript.

Kind regards,

Joy Sturtevant

Academic Editor

PLOS ONE

Journal Requirements:

Reviewers' comments:

Reviewer's Responses to Questions

**Comments to the Author**

1. Is the manuscript technically sound, and do the data support the conclusions?

Reviewer #1: Partly

Reviewer #2: Yes

2. Has the statistical analysis been performed appropriately and rigorously? 

Reviewer #1: Yes

Reviewer #2: Yes

3. Have the authors made all data underlying the findings in their manuscript fully available?

Reviewer #1: No

Reviewer #2: Yes

4. Is the manuscript presented in an intelligible fashion and written in standard English?

Reviewer #1: No

Reviewer #2: Yes

5. Review Comments to the Author

Reviewer #1: Although this paper could be potentially of interest, it suffers from including findings, which have been obviously and repeatedly reported elsewhere. Examples are below,

1. It is not clear why authors attempted to undertake the phenotypic assay, while antifungal susceptibility profile clearly showed that fluconazole at low doses was inhibitory for susceptible isolates, but not for resistant ones. Basically, this phenotypic assay is redundant and is suggested to be removed from the study.

2. Clinical profile of such isolates to be presented in an informative table, where authors include the following details, age/sex, unedrlying conditions, central venous catheter, corticosteroid and broadsectrum antibiotic use, mechanical ventilation, duration of hospitalization, antifungal treatment with doasge and duration, and finally outcome. These data are recommended to be used instead of the phenotypic analysis and the overall findings from this is encouraged to be discussed in discussion. Clearly most of the patients infected with azole resistant isolates are azole-naive and this finding should be discussed in the context of clinical data and that such infections were either acquired horizontally or from other unknown sources.

3. Instead of Table 1, authors can present a comprehensive table, where the MIC of each isolate along with amino acid substituted and also the fold expression of the genes studied can be shown. This way the data is more clear. Also please remove Figure 1.

4. The basis for choosing the genes studied by RT-qPCR has not been clarified. Have authors inferred this from RNAseq analysis of a specific fluconazole resistant C. tropicalis isolate? This needs to be clarified and clearly link their association with azole resistance to solidify the philosophy behind their inclusion in the current study.

5. Please explain which fluconazole dosage and duration was used to study the inducible expression of the target genes? This should be explicitly introduced in the results section.

6. Lines 240-247 do not deserve to be a separate heading and it should be a paragraph furthering the previous heading (line 208).

7. Sections dealing with homology modeling and docking analysis are not necessary given that this is a very well-known mutation, which confers azole resistance in various Candida species.

Reviewer #2: General comments

This study investigated the molecular mechanisms of azole resistance in Candida tropicalis, an emerging challenge in Asia, even globally. Like the resistant mechanisms in other Candida species, numbers of papers have illustrated several mechanisms involving ergosterol biosynthesis pathway or upregulation of efflux pumps in C. tropicalis. The findings in the present study not only echoed those in the previous research but also provided insight into the stress response pathways. The manuscript contains interesting data but also has some issues warranted further clarification with respect to methods and presentation.

Specific comments

1. The authors need to describe the methods for quantitative RT-PCR in details, especially the conditions of drug induction and the reference for calculation of relative gene expression in each strain. How many untreated control strains (Line 232) were used?

2. Based on 2-∆∆CT method (Line 232), all relative expression levels are theoretically above zero. But it’s unusual to depict negative values shown in the current figures and supplementary figures. Please check relative expression levels in each gene for all strains tested again. If indicated, please calculate the statistics among three groups (R-WM, R-WTM, and Susceptible) and revise the manuscript accordingly.

3. Among two nonsynonymous mutations (A395T and C461T) identified in the ERG11 gene, C461T dosen’t confer azole resistance per se by the site-direct mutagenesis experiment (reference 14). Please revise Line 360-361 accordingly.

4. In the current study, these two ERG11 mutations were simultaneously evaluated by the bioinformatics approaches. It’s interesting to test the impact of C461T on azole resistance by performing the homology modelling and molecular docking of single mutation.

5. Please provide the reference strain to determine the molecular alternations in ERG1, ERG3, UPC2, and TAC1 genes.

6. Please note that T503C in UPC2 gene has been reported among azole resistant C. tropicalis (https://doi.org/10.3390/jof7080612). Please revise Line 375-377 accordingly.

7. Throughout the manuscript, the gene names need to be italicized (Line 198 as an example), while protein names are not.

6. PLOS authors have the option to publish the peer review history of their article (what does this mean?). If published, this will include your full peer review and any attached files.

Reviewer #1: No

Reviewer #2: No

---

## [Author Response · Author response to Decision Letter 0]

24 Apr 2022

Editor comments:

Comment: The experimental parameters need to be stated and more clearly described.

Relative expression levels need to be readdressed as discussed by reviewer #2

Review the discussion and conclusions to confirm they reflect results (see reviewer #2) and also identify content that has already been reported (reviewer #1).

Please consider all the comments of the reviewers and if you think their suggestions would improve what you are trying to emphasize.

Response: Thank you for considering our manuscript in your esteemed journal. We have tried our best to address all the issues raised by the Academic Editor and Reviewers. We addressed the query of the second reviewer on the relative expression level. We modified the discussion and conclusion as per the suggestion of the second reviewer and also address the concerns of the first reviewer. 

Comment: Please include the following items when submitting your revised manuscript:

Response: We are submitting the rebuttal letter containing each point raised by the academic editor and reviewers.

We are also sending one copy of the ‘Revised Manuscript with Track Changes’ and one clean copy of the revised version labeled as ‘Manuscript’. 

Comment: Response: Required modifications in the financial disclosure has included in the updated statement of the cover letter. 

Comment: Guidelines for resubmitting your figure files are available below the reviewer comments at the end of this letter. 

Response: We are resubmitting the figure files according to the guidelines of this journal. 

Reviewers’ comments:    

Reviewer's Responses to Questions:

Comments:

1. Is the manuscript technically sound, and do the data support the conclusions? The manuscript must describe a technically sound piece of scientific research with data that supports the conclusions. Experiments must have been conducted rigorously, with appropriate controls, replication, and sample sizes. The conclusions must be drawn appropriately based on the data presented. 

Reviewer #1: Partly

Reviewer #2: Yes

2. Has the statistical analysis been performed appropriately and rigorously? 

Reviewer #1: Yes

Reviewer #2: Yes

3. Have the authors made all data underlying the findings in their manuscript fully available?The PLOS Data policy requires authors to make all data underlying the findings described in their manuscript fully available without restriction, with rare exception (please refer to the Data Availability Statement in the manuscript PDF file). The data should be provided as part of the manuscript or its supporting information, or deposited to a public repository. For example, in addition to summary statistics, the data points behind means, medians and variance measures should be available. If there are restrictions on publicly sharing data—e.g. participant privacy or use of data from a third party—those must be specified.

Reviewer #1: No

Reviewer #2: Yes

4. Is the manuscript presented in an intelligible fashion and written in standard English?PLOS ONE does not copyedit accepted manuscripts, so the language in submitted articles must be clear, correct, and unambiguous. Any typographical or grammatical errors should be corrected at revision, so please note any specific errors here.

Reviewer #1: No

Reviewer #2: Yes

Response: Thank you so much for your appreciation and positive responses.

Review Comments to the Author:

Reviewer #1

Although this paper could be potentially of interest, it suffers from including findings, which have been obviously and repeatedly reported elsewhere. Examples are below,

Comment 1: It is not clear why authors attempted to undertake the phenotypic assay, while antifungal susceptibility profile clearly showed that fluconazole at low doses was inhibitory for susceptible isolates, but not for resistant ones. Basically, this phenotypic assay is redundant and is suggested to be removed from the study.

Response: Thank you so much for your valuable suggestion. We removed the phenotypic assay from the revised manuscript. 

Comment 2: Clinical profile of such isolates to be presented in an informative table, where authors include the following details, age/sex, unedrlying conditions, central venous catheter, corticosteroid and broad-spectrum antibiotic use, mechanical ventilation, duration of hospitalization, antifungal treatment with doasge and duration, and finally outcome. These data are recommended to be used instead of the phenotypic analysis and the overall findings from this is encouraged to be discussed in discussion. Clearly most of the patients infected with azole resistant isolates are azole-naive and this finding should be discussed in the context of clinical data and that such infections were either acquired horizontally or from other unknown sources.

Response: Thank you so much for your suggestion. All the available clinical details have been added to table 1. However, we were unable to retrieve the complete clinical details of all patients. The clinical aspects not mentioned in the table 1 have discussed in the discussion section of the revised manuscript. 

Comment 3: Instead of Table 1, authors can present a comprehensive table, where the MIC of each isolate along with amino acid substituted and also the fold expression of the genes studied can be shown. This way the data is more clear. Also please remove Figure 1.

Response: Thank you so much for your meticulous comments. We are presenting the MIC of the isolates with amino acid substitution in Table 1 in the revised manuscript. We think the fold expression will not be best suited to the table format. Therefore, we are presenting the expression date in graph format in the figures. As per your suggestion, Figure 1 has been removed from the revised manuscript. 

Comment 4: The basis for choosing the genes studied by RT-qPCR has not been clarified. Have authors inferred this from RNAseq analysis of a specific fluconazole resistant C. tropicalis isolate? This needs to be clarified and clearly link their association with azole resistance to solidify the philosophy behind their inclusion in the current study.

Response: Thank you so much for your query. Unfortunately, we could not perform the RNAseq analysis due to the unavailability of adequate funds. The genes were selected based on the existing literature on azole resistance in different Candida species. The published articles on these genes are cited in this manuscript and also in our previously published article on the dynamics of in vitro development of azole resistance in Candida tropicalis (doi: 10.1016/j.jgar.2020.04.018). 

Comment 5: Please explain which fluconazole dosage and duration was used to study the inducible expression of the target genes? This should be explicitly introduced in the results section.

Response: As per your suggestion, we explained the fluconazole doses and duration of drug exposure to study the expression of target genes in the result section of the revised manuscript. 

Comment 6: Lines 240-247 do not deserve to be a separate heading and it should be a paragraph furthering the previous heading (line 208).

Response: We removed the heading for lines 240-247 and it is included as a paragraph with the previous heading. Relevant modifications have been made in the revised manuscript. 

Comment 7: Sections dealing with homology modeling and docking analysis are not necessary given that this is a very well-known mutation, which confers azole resistance in various Candida species.

Response: To the best of our knowledge, the exact role of the non-synonymous alteration in azole resistance has not been elucidated yet. It is known that tyrosine (Y) is an aromatic amino acid and is preferentially substituted by amino acids with similar properties and considering the fact that phenylalanine (F) is the precursor of tyrosine, it is a likely substitution in the ERG11. We confirmed for the first time that substitutions of Y132F and S154F hindered the formation of Pi-Pi and Pi-cation interaction between the cofactor and the ligand molecules thereby dipping the overall binding energy of the mutated docked complex. Therefore, for the present study homology modeling and docking analysis are innovative and help in understanding the effect of the mutations in drug binding, particularly in the context of C. tropicalis. With your permission, we wish to retain the homology modeling and docking data in the revised manuscript. 

Reviewer #2: 

This study investigated the molecular mechanisms of azole resistance in Candida tropicalis, an emerging challenge in Asia, even globally. Like the resistant mechanisms in other Candida species, numbers of papers have illustrated several mechanisms involving ergosterol biosynthesis pathway or upregulation of efflux pumps in C. tropicalis. The findings in the present study not only echoed those in the previous research but also provided insight into the stress response pathways. The manuscript contains interesting data but also has some issues warranted further clarification with respect to methods and presentation.

Comment 1: The authors need to describe the methods for quantitative RT-PCR in details, especially the conditions of drug induction and the reference for calculation of relative gene expression in each strain. How many untreated control strains (Line 232) were used?

Response: Thank you so much for your valuable suggestions. We described the method for RT-qPCR in detail in the method section of the revised manuscript.

 The conditions of drug induction are included in the result section of the modified manuscript.

 In our previously published study on the selection and evaluation of appropriate reference genes for RT-qPCR-based expression analysis in Candida tropicalis following azole treatment, we confirmed that EF1 is the most stable reference gene. Therefore, we used elongation factor 1α (EF1) as a reference gene for the calculation of relative gene expression in each strain.

 The drug-induced expression for each isolate was calculated with respect to the drug untreated control for the same isolate along with the reference gene. Therefore, the number of drug-treated samples is the same as the control. 

 Comment 2: Based on 2-ΔΔCT method (Line 232), all relative expression levels are theoretically above zero. But it’s unusual to depict negative values shown in the current figures and supplementary figures. Please check relative expression levels in each gene for all strains tested again. If indicated, please calculate the statistics among three groups (R-WM, R-WTM, and Susceptible) and revise the manuscript accordingly.

Response: We calculated the gene expression using 2-ΔΔCT method based on the study entitled “Analyzing real-time PCR data by the comparative CT method” by Thomas D Schmittgen & Kenneth J Livak (doi:10.1038/nprot.2008.73). According to this study, if the CT for the treated sample is higher than the untreated sample, the level of gene expression is -1/calculated fold change. The result will be negative and which confirms that the expression is reduced due to treatment. In our study, the negative values shown in the current figures and supplementary figures confirm that drug treatment decreased the expression of the genes when compared with their untreated control. There according to us, the calculation for the gene expression is perfect and flawless. 

Comment 3: Among two nonsynonymous mutations (A395T and C461T) identified in the ERG11 gene, C461T doesn’t confer azole resistance per se by the site-direct mutagenesis experiment (reference 14). Please revise Line 360-361 accordingly.

Response: Thank you so much for your valuable suggestion. Relevant changes have been made in the revised manuscript.

Comment 4: In the current study, these two ERG11 mutations were simultaneously evaluated by the bioinformatics approaches. It’s interesting to test the impact of C461T on azole resistance by performing the homology modelling and molecular docking of single mutation.

Response: As per your suggestion, we performed the homology modelling and molecular docking of the C461T single mutation and have presented the results for your reference below. However, since the results do not seem to add much to the current manuscript we have not added them to the revised draft.

 The figures showed that Serine/Phenylalanine at 154 position does not interact with the Heme and any of the 2 drugs. Indicating that the amino acids are not associated with drug interaction unlike Tyrosine 132 position which forms a bond with the drugs. Both Serine and Phenylalanine form polar bonds with their neighboring amino acids and the number of bonds decreases as Serine mutates to Phenylalanine from 7 to 4.

A) The superimposed S154F mutation with native structure in a Fluconazole and protein complex. Tyrosin 132 (green), Heme (red), Fluconazole (Blue), S154 (orange) superimposed by F154 (purple). The S154F does not interact with the fluconazole heme Tyrosine 132 complex.

A.1. The polar bond among the Serine 154 (orange) residues and its neighboring amino acids (purple) indicated by yellow dotted lines, while the ligands heme (red) and Fluconazole (blue) are in polar bond with Tyrosine (cyan) and neighboring amino acids (purple) and A.2. Polar bonds between mutated phenylalanine 154 residue (green) in polar contact with neighbouring amino acids and no change in the polar bonds between Heme, Fluconazol and Tyrosine 132

 A.1 A.2

B) Fig shows the superimposed S154F mutation with native structure in a Voriconazole and protein complex Tyrosin 132 (cyan), Heme (red), Voriconazole (Blue) , S154 (orange) superimposed by F154 (green)

B.1. Fig shows the polar bond among the Serine 154 residues (green) and the neighboring amino acids. The heme and voriconazole interact with Tyrosine 132(cyan) independent of Serine and B.2. Polar interaction between phenylalanine (light blue) and the neighboring amino acids (in purple).

 B.1 B.2

Comment 5: Please provide the reference strain to determine the molecular alternations in ERG1, ERG3, UPC2, and TAC1 genes.

Response: The details of the reference strain have been included in the modified manuscript. 

Comment 6: Please note that T503C in UPC2 gene has been reported among azole resistant C. tropicalis (https://doi.org/10.3390/jof7080612). Please revise Line 375-377 accordingly.

Response: We agree with you and the lines have been modified in the revised manuscript.

Comment 7: Throughout the manuscript, the gene names need to be italicized (Line 198 as an example), while protein names are not.

Response: The gene and protein names are corrected as per the suggestion and relevant changes have been included in the modified and revised manuscript.

---

## [Decision Letter · Decision Letter 1]

27 May 2022

Mechanisms of azole antifungal resistance in clinical isolates of Candida tropicalis

PONE-D-21-29687R1

Dear Dr. Ghosh,

We’re pleased to inform you that your manuscript has been judged scientifically suitable for publication and will be formally accepted for publication once it meets all outstanding technical requirements.

Kind regards,

Joy Sturtevant

Academic Editor

PLOS ONE

Additional Editor Comments (optional):

Reviewers' comments:

Reviewer's Responses to Questions

**Comments to the Author**

1. If the authors have adequately addressed your comments raised in a previous round of review and you feel that this manuscript is now acceptable for publication, you may indicate that here to bypass the “Comments to the Author” section, enter your conflict of interest statement in the “Confidential to Editor” section, and submit your "Accept" recommendation.

Reviewer #2: All comments have been addressed

2. Is the manuscript technically sound, and do the data support the conclusions?

Reviewer #2: Yes

3. Has the statistical analysis been performed appropriately and rigorously? 

Reviewer #2: Yes

4. Have the authors made all data underlying the findings in their manuscript fully available?

Reviewer #2: Yes

5. Is the manuscript presented in an intelligible fashion and written in standard English?

Reviewer #2: Yes

6. Review Comments to the Author

Reviewer #2: As shown in example 1 of the reference 27 (10.1038/nprot.2008.73), the value of 2-ΔΔCT<1 implies that there was a reduction in the expression due to treatment. If the authors choose to present the level of gene expression as -1/calculated fold change for which CT for the treated sample is higher than the untreated sample, it would be better to describe briefly regarding this methodology in the method section.

7. PLOS authors have the option to publish the peer review history of their article (what does this mean?). If published, this will include your full peer review and any attached files.

Reviewer #2: No

---

## [Editor Report · Acceptance letter]

4 Jul 2022

PONE-D-21-29687R1 

Mechanisms of azole antifungal resistance in clinical isolates of *Candida tropicalis*

Dear Dr. Ghosh:

I'm pleased to inform you that your manuscript has been deemed suitable for publication in PLOS ONE. Congratulations! Your manuscript is now with our production department. 

Kind regards, 

on behalf of

Dr. Joy Sturtevant 

Academic Editor

PLOS ONE